# STRUCTURING HIDDEN FEATURES VIA CLUSTERING OF UNIT-LEVEL ACTIVATION PATTERNS

## ABSTRACT

We propose a self-supervised learning framework that organizes hidden feature representations across layers, thereby enhancing interpretability. The framework first discovers unit-level structures by comparing activation patterns across data samples. Building on these structures, we introduce a structure-aware regularization objective that (i) promotes feature reuse across layers via identity mappings and (ii) encourages the emergence of representative units that serve as anchors for related features. This regularization yields clearer and more structured feature pathways, enhancing the interpretability of the learned representations. Experiments demonstrate that our method induces structured feature pathways on synthetic data, improves interpretability on CIFAR-10 as measured by Grad-CAM++ metrics, and maintains competitive performance with slightly improved mean accuracy on both CIFAR-10 and ImageNet-1K.

## 1 INTRODUCTION

Deep neural networks learn complex internal feature representations that often lack explicit structure, leading to inefficient training and reduced interpretability. In particular, similar features are frequently re-learned across layers rather than reused, and features compete inefficiently across and within layers, which obscures their semantic roles. To address these inefficiencies, we propose a self-supervised learning framework that structures hidden features by identifying and regularizing activation-level similarities across the model. Our method introduces a clustering-based analysis of hidden units, along with a regularization loss that promote cross-layer feature reuse through identity mapping and encourage competition centered around a representative anchor. As shown in Figure 1, our approach yields more compact and semantically structured representations, promoting inter-layer feature reuse through residual connections while facilitating feature exploration centered around the layer containing anchor features.

Existing studies have also attempted to analyze or enhance feature representations, for example by modifying representations or designing new architectures. However, these methods typically operate on layer-level features, which limits their granularity and makes it difficult to capture fine-grained relationships between hidden units. In addition, many of them rely on architectural modifications or auxiliary modules.

In contrast, we introduce an architecture-agnostic strategy that operates directly at the hidden-unit level. To operate at this level, we define each hidden feature as the activation of a single unit across multiple input samples. We then cluster these features based on similarities in their rank-transformed activation patterns, which reduces sensitivity to absolute magnitudes and noisy fluctuations in the activations. This reveals structural patterns, such as inefficient regeneration across layers and the exploration of features throughout the model.

To guide learning based on these observations, we design a novel self-supervised objective. The *structure loss* organizes cross-layer feature reuse patterns by aligning features at residual positions within the same cluster, effectively reducing inefficient regeneration and promoting the emergence of a representative anchor feature in each cluster—thereby fostering structured competition.

Our method integrates seamlessly into standard training pipelines without requiring architectural modifications. Leveraging this property, we apply it to Vision Transformers (ViTs) and evaluate induced structured internal representations, model interpretability, and downstream task performance.

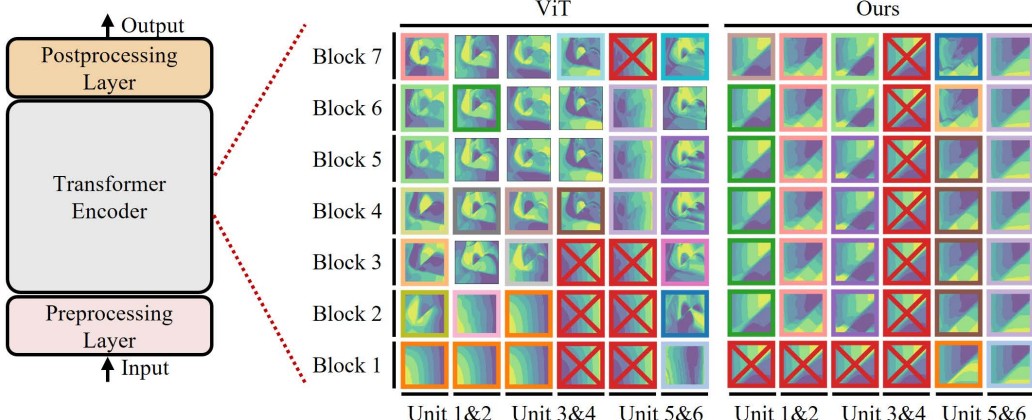

Figure 1: Visualization of hidden features in a Vision Transformer (ViT) for six sampled units across the first seven transformer blocks, comparing the baseline ViT (middle) and our method (right), using the synthetic dataset. Each column group (e.g., Unit 1) corresponds to a fixed (token, embedding dimension) index, and each row represents a different block. Features are shown as contour plots, with cluster membership indicated by color-coded boundaries. For clarity, 20 clusters are highlighted with bold colored borders in distinct hues, while all other clusters are outlined with thin black borders. Features belonging to a specific cluster are additionally marked with bold red Xs, which, in our method, correspond to concentrated feature exploration in the earliest layer and effective feature reuse via identity mapping through residual connections.

On a synthetic dataset, it yields better-structured representations, characterized by the emergence of representative units in the earliest layer—serving as anchors for competitive exploration of related features—and by effective feature reuse via identity mapping through residual connections. On CIFAR-10, our approach produces more focused attribution maps, demonstrating improved explainability. On CIFAR-10 and ImageNet-1K, it achieves slightly improved accuracy. In summary, our framework organizes hidden features into structured representations that enhance internal organization and interpretability while maintaining comparable performance.

## 2 RELATED WORK

Our method addresses both analyzing internal representations and improving feature structure by clustering hidden units via rank-transformed activation patterns, capturing fine-grained similarity without altering the model or inputs. In contrast, prior work often relies on layer-level representations or requires architectural changes or external modules, as discussed in the following two categories.

**Analyzing Internal Representations.** Understanding hidden features is key to interpreting and improving deep neural networks. Some methods assess feature importance by modifying or masking activations (Bandarkar et al., 2025; Feng et al., 2024; Kim et al., 2025; Jiang et al., 2024), but such interventions can distort activation distributions. Others compare feature similarity via correlation-based metrics (Song et al., 2025; Dravid et al., 2023; Huh et al., 2024) or apply clustering to low-dimensional projections (Donahue et al., 2014), yet these often rely only on layer-level representations, limiting granularity, or require access to multiple pretrained models.

**Improving Feature Structure.** Recent work has enhanced internal feature relationships by adding alignment objectives (Wang et al., 2023; Lee et al., 2023; Kim et al., 2023), often requiring extra components such as temporal alignment modules or auxiliary classifiers. Others modify the architecture (Guo & Gan, 2024; Xia et al., 2024; Zhai et al., 2023), for example with convolutional fusion modules to improve feature interactions.

---

**Algorithm 1** Overview of our algorithm

---

1: $\mathcal{C} \leftarrow \emptyset$
2: $l_{\text{main}}, l_{\text{structure}} \leftarrow 0, [\,]$
3: $F \leftarrow \text{Sampling}()$          ▷ Section 3.1
4: **for** $t = 1$ to $T$ **do**
5:      $(x, y) \sim D$
6:      $(\hat{y}, F, f) \leftarrow \text{Forward}(\theta, x, F)$          ▷ Section 3.2
7:      $l_{\text{main}} \leftarrow l_{\text{main}} + L_{\text{main}}(\hat{y}, y)$
8:      $l_{\text{structure}} \leftarrow \text{StructureLoss}(l_{\text{structure}}, f, \mathcal{C})$          ▷ Section 3.4
9:      **if** $t \bmod B_{\text{train}} = 0$ **then**
10:          $l_{\text{total}} \leftarrow \text{WeightedMean}(l_{\text{main}}, l_{\text{structure}}, B_{\text{train}})$          ▷ Section 3.5
11:          $\theta \leftarrow \text{Backpropagation}(\theta, l_{\text{total}})$
12:          $l_{\text{main}}, l_{\text{structure}} \leftarrow 0, [0 \,|\, c \in \mathcal{C}, \; i \in \mathcal{I}_c \setminus \{\min(c)\}]$
13:      **end if**
14:      **if** $t \bmod B_{\text{cluster}} = 0$ **then**
15:          $\mathcal{C} \leftarrow \text{Clustering}(F)$          ▷ Section 3.3
16:          $F \leftarrow \text{Sampling}()$
17:          $l_{\text{structure}} \leftarrow [0 \,|\, c \in \mathcal{C}, \; i \in \mathcal{I}_c \setminus \{\min(c)\}]$
18:      **end if**
19: **end for**
20: **return** $\theta$

---

## 3 METHOD

As outlined in Algorithm 1, we structure hidden features by clustering unit-level activation patterns and guiding them with a self-supervised objective, structure loss. This loss encourages cross-layer feature reuse via residual connections and organizes similar features within a layer around a representative anchor unit. Our approach improves the efficiency and organization of hidden representations without altering the network architecture.

In detail, we extend the standard training loop with five additional components. First, we periodically sample a shared subset of consecutive layer and token indices to control the computational cost overhead introduced by our algorithm, as described in Section 3.1. Second, we extract hidden activations from the sampled layers and token indices during the forward pass, as described in Section 3.2. Third, we identify structural patterns in the collected hidden features by transforming their activation values into ranks over data samples, and clustering them based on whether their pairwise similarity falls below a threshold, as described in Section 3.3. Fourth, we compute the structure loss, which encourages the emergence of a representative anchor unit—enhancing cross-layer reuse via identity mapping in residual connections and fostering competitive exploration around the anchor unit—as described in Section 3.4. Finally, we balance the proposed loss with the original main loss, as described in Section 3.5.

### 3.1 SAMPLING HIDDEN FEATURE INDICES

To control the computational and memory overhead of applying our algorithm to hidden activations, we periodically sample a subset of hidden activation indices based on a predefined clustering interval $B_{\text{cluster}}$. At each interval, we randomly select $S_{\text{layer}}$ consecutive layers and $S_{\text{token}}$ consecutive token positions. Formally, if the total number of hidden activations in the model is

$$H_{\text{whole}} = O_{\text{layer}} \times O_{\text{token}} \times O_{\text{embed}},$$

where $O_{\text{layer}}$, $O_{\text{token}}$, and $O_{\text{embed}}$ denote the total number of layers, the total number of token positions, and the embedding dimension, respectively, then our sampling process extracts

$$H_{\text{select}} = S_{\text{layer}} \times S_{\text{token}} \times O_{\text{embed}}.$$

This design controls the complexity by operating only on a sampled set of hidden activations, which scales as $\mathcal{O}(H_{\text{select}})$, with a maximum of $\mathcal{O}(H_{\text{whole}})$ if all activations were used. As a result, the per-sample feature $f \in \mathbb{R}^{H_{\text{select}}}$ and the aggregated features $F \in \mathbb{R}^{B_{\text{cluster}} \times H_{\text{select}}}$ are employed to identify and structure the hidden features. Moreover, by periodically refreshing the sampled indices throughout training, the method could gradually achieve full-feature coverage across the model.

$$
\begin{bmatrix} 0.81 & 0.87 & ... & -0.09 & -0.51 \end{bmatrix}
$$

$$
=
$$

$$
\begin{bmatrix}
0.81 & 0.87 & -0.32 & 0.09 \\
-0.22 & -0.85 & -0.50 & -0.48 \\
0.99 & -0.62 & 0.80 & -0.19 \\
-0.36 & 0.65 & -0.09 & -0.51
\end{bmatrix}
$$

$$
\begin{bmatrix} 13 & 14 & ... & 9 & 2 \end{bmatrix}
$$

$$
=
$$

$$
\begin{bmatrix}
13 & 14 & 6 & 10 \\
7 & 0 & 3 & 4 \\
15 & 1 & 12 & 8 \\
5 & 11 & 9 & 2
\end{bmatrix}
$$

$$
\begin{bmatrix} 3 & 3 & ... & 2 & 0 \end{bmatrix}
$$

$$
=
$$

$$
\begin{bmatrix}
3 & 3 & 1 & 2 \\
1 & 0 & 0 & 1 \\
3 & 0 & 3 & 2 \\
1 & 2 & 2 & 0
\end{bmatrix}
$$

**Feature (Raw)**
Activation values
for input samples
$(B_{\mathrm{cluster}} : 16)$

**Feature (Rank)**
Ranking values

**Feature (Grouped Rank)**
Number of bins
$(\beta : 4)$

Figure 2: Feature transformation during identifying structures. Raw activation values $F_{:,h_{\mathrm{select}}}$ are converted into grouped ranks $\hat{F}_{:,h_{\mathrm{select}}}$ to reduce sensitivity to magnitude and noise. The example shows $B_{\mathrm{cluster}} = 16$ samples divided into $\beta = 4$ bins. For visualization, the 1D feature vector is reshaped into a 2D grid using the square root of $B_{\mathrm{cluster}}$ to make the layout more compact.

## 3.2 Extracting Hidden Features

During training, we compute the main task loss from predictions while recording selected unit activations as per-sample features $f$. Over a clustering interval $B_{\mathrm{cluster}}$, these are aggregated into $F$ for identifying structures, while $f$ directly informs the structure loss to organize hidden representations.

## 3.3 Identifying Structures via Unit-Level Clustering

To uncover structural patterns among the extracted hidden features, we perform clustering over the aggregated features $F$ using a rank-based similarity measure. The process consists of three stages: grouped rank transformation, distance computation, and graph-based clustering.

**Grouped Rank Transformation.** To mitigate the sensitivity of feature comparisons to absolute magnitude differences and noise in features, we apply a grouped rank transformation to each hidden unit, as in Figure 2. For a hidden unit represented by the feature vector $F_{:,h_{\mathrm{select}}} \in \mathbb{R}^{B_{\mathrm{cluster}}}$, we sort activation values across the $B_{\mathrm{cluster}}$ samples and divide them into a fixed number of bins $\beta$. Each bin corresponds to a rank range, and all values within the same bin share a rank index. This produces the transformed feature $\hat{F} \in \mathbb{R}^{B_{\mathrm{cluster}} \times H_{\mathrm{select}}}$:

$$
\hat{F}_{:,h_{\mathrm{select}}} = \left\lfloor \frac{\mathrm{rank}(F_{:,h_{\mathrm{select}}}) \, \beta}{B_{\mathrm{cluster}}} \right\rfloor,
$$

where $\mathrm{rank}(\cdot) \in \{0, \ldots, B_{\mathrm{cluster}} - 1\}$ denotes the 0-based rank after sorting. This transformation standardizes activation scales while preserving the relative ordering of activations across samples.

**Distance Computation.** To obtain structure information related to feature reuse, we compute pairwise L1 distances between transformed features to form a distance matrix $M \in \mathbb{R}^{H_{\mathrm{select}} \times H_{\mathrm{select}}}$:

$$
M_{h_{\mathrm{select}}, h'_{\mathrm{select}}} = \left\| \hat{F}_{:,h_{\mathrm{select}}} - \hat{F}_{:,h'_{\mathrm{select}}} \right\|_1,
$$

where $h_{\mathrm{select}}$ and $h'_{\mathrm{select}}$ denote two sampled units with their respective indices.

**Graph-Based Clustering.** To cluster based on computed distances, we construct a graph using a difference threshold $\tau$. The binary adjacency matrix $A \in \{0, 1\}^{H_{\mathrm{select}} \times H_{\mathrm{select}}}$ is defined as

$$
A = \mathbf{1}_{M < \tau},
$$

where $\mathbf{1}_{M < \tau}$ denotes the element-wise indicator function, i.e., it returns 1 for each entry $M_{h_{\mathrm{select}}, h'_{\mathrm{select}}}$ when $M_{h_{\mathrm{select}}, h'_{\mathrm{select}}} < \tau$ holds, and 0 otherwise. We set $\tau = B_{\mathrm{cluster}}$, which corresponds to allowing at most an average group rank difference of 1 per sample. The resulting undirected graph connects highly similar units, and its connected components form clusters $\mathcal{C} = \{c\}$. We then retain only clusters in $\mathcal{C}$ that contain at least two units.

### 3.4 STRUCTURING FEATURES VIA SELF-SUPERVISED LOSS

With the structure information obtained from clustering, we introduce a self-supervised structure loss. Its purpose is twofold: to prevent information loss during cross-layer feature reuse by promoting identity mapping, and to reduce inefficient exploration caused by excessive competition among features across multiple layers. This is achieved by encouraging the reuse of a representative anchor unit over residual connections within each cluster.

In detail, for each cluster $c \in \mathcal{C}$, the unit at the first index is designated as the representative anchor unit. Alignment is then enforced only for units that share both the token position $s^*_{\text{token}}$ and the embedding dimension $o^*_{\text{embed}}$ with this anchor, ensuring its reuse via residual connections across layers within the cluster. The set of layers in cluster $c$ is determined by the minimum and maximum layer indices, denoted by $\min(c)$ and $\max(c)$, respectively. Formally, the set of layer indices for cluster $c$ is defined as

$$\mathcal{I}_c = \{i \in \mathbb{Z} \mid \min(c) \leq i \leq \max(c)\}.$$

For all clusters $c \in \mathcal{C}$ and for all layer indices $i \in \mathcal{I}_c \setminus \{\min(c)\}$—that is, all layers in the cluster except the one containing the anchor unit—we compute an alignment loss with the per-sample feature $f$ and the anchor index $(\min(c), s^*_{\text{token}}, o^*_{\text{embed}})$ of the cluster $c$

$$L_{\text{structure}}(f, c, i) = \left\| f_{(i,\, s^*_{\text{token}}, o^*_{\text{embed}})} - \text{stopgrad}\left( f_{(\min(c),\, s^*_{\text{token}}, o^*_{\text{embed}})} \right) \right\|_2^2,$$

which measures how closely each unit follows the anchor. The function $\text{stopgrad}(\cdot)$ prevents gradient flow into the anchor unit so that it remains a fixed reference during optimization.

The result, list of all alignment losses across clusters, is given by:

$$\left[ \ell_{c,i} \leftarrow \ell_{c,i} + L_{\text{structure}}(f, c, i) \mid c \in \mathcal{C},\ i \in \mathcal{I}_c \setminus \{\min(c)\} \right],$$

which means these per-sample alignment losses are collected and accumulated over each mini-batch of size $B_{\text{train}}$ during training.

### 3.5 OTHER SUPPORTING PROCEDURES

To stabilize training and prevent the structure loss from taking precedence over the main task optimization, we apply a filtering and weighting scheme to alignment losses before aggregating it into the total loss $l_{\text{total}}$. Specifically, for a given training batch of size $B_{\text{train}}$, we compare the mean structure loss $\bar{l}_{\text{structure}}$ to the averaged main task loss $\bar{l}_{\text{main}}$,

$$\bar{l}_{\text{main}} = \frac{l_{\text{main}}}{B_{\text{train}}}, \qquad \bar{l}_{\text{structure}} = \left[ \bar{\ell}_{c,i} = \frac{\ell_{c,i}}{B_{\text{train}}} \,\middle|\, c \in \mathcal{C},\ i \in \mathcal{I}_c \setminus \{\min(c)\} \right].$$

Any averaged alignment loss exceeding the averaged main task loss is discarded from the aggregation, ensuring that the auxiliary objective does not overshadow the primary optimization signal. Formally, the aggregated structure loss is computed as

$$l_{\text{aggregated}} = \frac{\displaystyle\sum_{\bar{\ell}_{c,i} \in \bar{l}_{\text{structure}}} \mathbf{1}_{\bar{\ell}_{c,i} \leq \bar{l}_{\text{main}}} \cdot \bar{\ell}_{c,i}}{\displaystyle\sum_{\bar{\ell}_{c,i} \in \bar{l}_{\text{structure}}} \mathbf{1}_{\bar{\ell}_{c,i} \leq \bar{l}_{\text{main}}}},$$

where $\mathbf{1}_{\bar{\ell}_{c,i} \leq \bar{l}_{\text{main}}}$ is the indicator function and the denominator counts the number of retained losses to produce a mean value over the surviving set. The final training objective is then defined as:

$$l_{\text{total}} = \bar{l}_{\text{main}} + \gamma\, l_{\text{aggregated}},$$

where $\gamma$ is a hyperparameter controlling the relative contribution of the structure loss. Model parameters are updated via standard backpropagation with respect to $l_{\text{total}}$.

## 4 EXPERIMENTS

We evaluate our proposed framework across three core aspects: **(i)** its ability to induce structured internal feature representations, **(ii)** its impact on model interpretability, and **(iii)** its effect on downstream task performance.

We first investigate *structured hidden features* using a synthetic dataset, where the low-dimensional nature of the task facilitates clear visualization of unit-level activation patterns and cross-layer feature reuse. Next, we evaluate *model interpretability* on CIFAR-10 using Grad-CAM++ visualizations, highlighting systematic changes in attribution maps induced by our method compared to a baseline. Finally, we assess *task performance* on CIFAR-10 and ImageNet-1K, showing that our approach consistently improves interpretability without compromising classification accuracy.

## 4.1 PREREQUISITE

A brief overview of the datasets and hyperparameters is provided below, while a complete description of all factors, including hardware specifications, is provided in Appendix B.

**Datasets.** We evaluate our framework on three datasets of varying complexity and scale: a synthetic dataset, CIFAR-10 (Krizhevsky & Hinton, 2009), and ImageNet-1K (Russakovsky et al., 2015). The synthetic dataset, inspired by the TensorFlow Playground (Hoeiness et al., 2021), is designed for fine-grained inspection of internal representation structures. It consists of two-dimensional binary classification tasks generated via a spiral function, with a dense grid of points for testing to facilitate visualization. CIFAR-10 is used to evaluate both classification performance and interpretability in a real-world image classification setting. ImageNet-1K serves as a large-scale benchmark to assess the scalability of our method.

**Hyperparameters.** We group all hyperparameters into four categories: *model-related*, *dataset-related*, *training-related*, and *method-related*. Our default experimental setting employs the Vision Transformer (ViT) architecture (Dosovitskiy et al., 2021) with a token embedding dimension of 256 and 14 transformer layers, while for the synthetic dataset we reduce the patch size to $1 \times 1$ to match its low-dimensional inputs, and for ImageNet-1K we use a larger ViT variant with short-epoch training. Dataset preprocessing follows the default configuration in the PyTorch Image Models library (Wightman, 2019), with geometric and color augmentations disabled for the synthetic dataset.

## 4.2 EVALUATION METRICS: STRUCTURED INTERNAL FEATURE REPRESENTATIONS

We assess the organization of internal representations using three categories of metrics: visualization of features and clusters, per-layer analysis, and per-cluster analysis.

**Visualization of Features and Clusters.** To visualize feature grouping and spatial organization, each selected group transformed feature $\hat{F}$ is reshaped into a 2D grid by using the square root of the number of samples in the vector ($B_{\text{cluster}}$), producing a compact layout, as explained in Figure 2. The resulting 2D features are then visualized as contour plots, with color-coded boundaries indicating cluster membership. Figure 1 presents multiple feature visualizations arranged according to their corresponding layers, enabling direct comparison of cluster structures across network depth.

**Per-layer Analysis.** We evaluate the diversity of features using a per-layer analysis. First, to assess inter-layer diversity, we leverage the distance matrix $M$ obtained during the identifying structures process in Section 3.3. We then apply min–max normalization and map it to grayscale intensities, yielding $\hat{M} \in \mathbb{R}^{H_{\text{select}} \times H_{\text{select}}}$:

$$\hat{M}_{h_{\text{select}}, h'_{\text{select}}} = 255 \cdot \frac{M_{h_{\text{select}}, h'_{\text{select}}} - \min(M)}{\max(M) - \min(M)},$$

where brighter regions correspond to greater dissimilarity at the unit level.

For layer-level difference analysis, we reshape $\hat{M}$ into $\mathbb{R}^{S_{\text{layer}} \times (S_{\text{token}} \cdot O_{\text{embed}} \cdot S_{\text{layer}} \cdot S_{\text{token}} \cdot O_{\text{embed}})}$, explicitly separating the first layer index from the remaining dimensions. We then average over all dimensions except the first layer index, yielding $\bar{M} \in \mathbb{R}^{S_{\text{layer}}}$, where each entry represents the average feature difference associated with that layer.

Second, to assess intra-layer diversity, we count the number of distinct non-overlapping clusters containing features from each layer, which represents the number of distinct feature groups within that layer.

**Per-cluster Analysis.** We perform per-cluster analysis to evaluate information preservation through identity mapping and the efficient exploration around anchor features.

To measure information preservation, we identify clusters that exhibit non-contiguous feature usage across layers, indicating that regeneration with potential information loss occurs instead of identity mapping over residual connections. This is done by extracting the unique layer indices for each cluster. A cluster is considered inefficient if

$$\max(c) - \min(c) + 1 > \# \text{ unique layers in the cluster},$$

which implies that similar features are reused while skipping intermediate layers.

To assess efficient exploration around anchor features, we construct a histogram of cluster sizes using fixed-width bins (size 10, range 0–200) and compare their frequencies on a logarithmic scale to capture both small and large clusters. We also report the size of the largest cluster as an indicator of the strength of dominant feature exploration.

### 4.3 Evaluation Metrics: Model Interpretability

We evaluate model interpretability using three metrics: visualization, Point Game (PG), and Energy PG. For all three metrics, we employ Grad-CAM++ (Chattopadhay et al., 2018), a gradient-based explanation method that highlights regions of the input image contributing most strongly to the model's decision. Specifically, Grad-CAM++ is applied to the output features of the final transformer block, enabling us to observe how structured representations influence semantic focus and spatial attribution within the input.

PG and Energy PG are segmentation-based metrics, adapted from their bounding-box counterparts (Chen et al., 2025) by replacing bounding boxes with segmentation masks generated by the Segment Anything Model (Kirillov et al., 2023). These masks are then used to evaluate the spatial alignment between predicted class activation maps and the locations of visually salient objects in the image.

**Visualization.** We generate visual explanations by overlaying Grad-CAM++ heatmaps (hereafter referred to as CAMs) on the original RGB input images. Each CAM is min–max normalized, resized to match the image resolution, and mapped to a blue-to-red colormap, where blue indicates low relevance and red indicates high relevance to the predicted class. The heatmap is then blended with the original image using an image weight of 0.7, preserving scene context while clearly highlighting salient regions. These overlays facilitate intuitive interpretation of which parts of the image most strongly influence the model's decision.

**Point Game.** The PG metric evaluates whether the most salient spatial location of the CAM lies within the predicted segmentation mask. A binary mask $\mathbf{G}_n$ is constructed with a single 1 at the position of the maximum CAM value and 0 elsewhere, and then flattened into a vector for subsequent computations. The metric is defined as

$$\frac{1}{N} \sum_{n=1}^{N} \langle \mathbf{G}_n, \mathbf{S}_n \rangle,$$

where $\mathbf{S}_n$ denotes the flattened binary segmentation mask for the $n$-th image and $N$ is the total number of images. This value represents the fraction of samples whose global CAM maximum lies inside the segmentation mask.

**Energy PG.** The Energy PG metric quantifies the spatial alignment between CAMs and the predicted segmentation masks by measuring the proportion of total CAM activation energy that lies within the segmentation region. It is defined as

$$\frac{1}{N} \sum_{n=1}^{N} \frac{\langle \mathbf{P}_n, \mathbf{S}_n \rangle}{\langle \mathbf{P}_n, \mathbf{1} \rangle + \epsilon},$$

where $\mathbf{P}_n$ denotes the flattened grayscale CAM for the $n$-th image, $\mathbf{S}_n$ denotes the flattened binary segmentation mask, $N$ is the total number of images, and $\epsilon$ is a small positive constant (set to $10^{-8}$) to avoid division by zero.

Table 1: Comparison of the proposed framework with the baseline ViT across three datasets. For the synthetic dataset (each layer contains 768 units), we report per-layer differences, per-layer cluster counts, the number of inefficient clusters, and the size of the largest cluster. For CIFAR-10, we report Point Game, Energy PG, and classification accuracy (percentages). For ImageNet-1K, we report top-1 classification accuracy (percentage) using a larger model under short-epoch training. Arrows indicate whether higher (↑) or lower (↓) values correspond to better performance. **Bold values** in the table indicate the better result. Statistical significance analyses are provided in Appendix C.

| Synthetic Layer (Block) | Difference (↑) ViT | Ours | Count (↓) ViT | Ours | Dataset/Metrics | ViT | Ours |
|---|---|---|---|---|---|---|---|
| 1 | **166.97** | 162.32 | 585 | **377** | | | |
| 2 | **163.61** | 161.83 | 733 | **698** | Synthetic | | |
| 3 | 161.64 | **161.85** | 762 | **704** | Inefficient clusters (↓) | 8 | **4** |
| 4 | 160.66 | **161.89** | 766 | **709** | Largest cluster (↑) | 94 | **179** |
| 5 | 159.54 | **161.94** | 766 | **722** | | | |
| 6 | 157.51 | **162.17** | 766 | **734** | CIFAR-10 | | |
| 7 | 156.02 | **162.30** | 767 | **740** | Point Game (↑) | 61.25 | **70.28** |
| 8 | 155.44 | **162.42** | 767 | **740** | Energy PG (↑) | 48.70 | **64.35** |
| 9 | 153.43 | **162.43** | 766 | **748** | Acc. (↑) | 97.49 | **97.58** |
| 10 | 151.12 | **162.58** | 755 | **753** | | | |
| 11 | 150.80 | **162.76** | **753** | 763 | ImageNet-1K | | |
| 12 | 150.45 | **163.59** | **753** | 768 | Acc. (↑) | 65.00 | **65.12** |
| 13 | 150.19 | **162.74** | 752 | **539** | | | |
| 14 | 148.95 | **162.85** | 749 | **558** | | | |

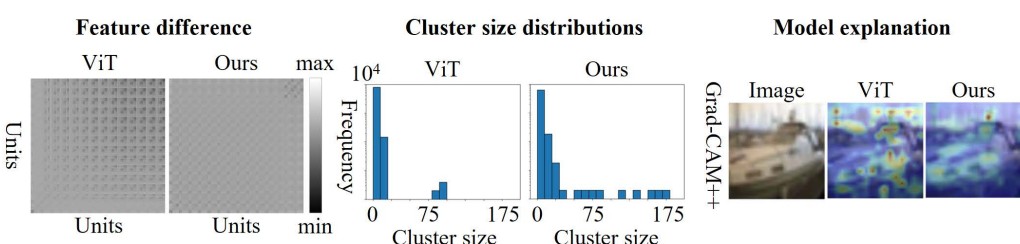

Figure 3: Visual comparison of structured feature organization and model explanations. We show from left to right: unit-level feature difference matrix $\hat{M}$ (synthetic dataset; each axis denotes hidden features ordered by layer, token, and embedding dimension, starting from the lower-left corner), cluster size distribution (synthetic dataset), and Grad-CAM++ heatmaps (CIFAR-10).

## 4.4 RESULTS

We evaluate structured internal feature representations through visualization of features and clusters, per-layer analysis, and per-cluster analysis, demonstrating that similar features are predominantly explored within the same layer and propagate efficiently via residual connections. Furthermore, model interpretability metrics reveal sharper and more class-focused explanations. Finally, down-stream task performance show that our framework slightly improves the baseline, confirming that these interpretability gains are achieved without sacrificing accuracy.

**Visualization of Features and Clusters.** Visualizations (Figure 1) show that in the baseline ViT, features from the same cluster (marked with "X") are inefficiently explored across multiple layers. For example, in blocks 1–3, multiple units such as unit 4–5 are both marked with "X," indicating that exploration of these features occurs across several layers. Moreover, for unit 4, the same feature disappears for blocks 4–6, and then reappears in block 7, illustrating reuse across non-adjacent layers. In contrast, our method shows feature exploration in block 1 primarily through units 1–4, and for unit 4, the same feature is reused continuously from block 1 through block 7 without interruption.

**Per-layer Analysis.** Per-layer analysis of the average per-unit difference (Figure 3) and per-layer difference values (Table 1) shows that our method maintains higher inter-layer diversity than the baseline in deeper layers. Our differences remain above 160 for all layers, whereas the baseline declines steadily, falling below 150 in the final layer. In contrast, per-layer cluster counts (Table 1) are consistently lower in our method, e.g., from 585 to 377 in the first layer and from 762 to 704 in the third layer, indicating more concentrated exploration within each layer.

**Per-cluster Analysis.** From the per-cluster perspective, the number of inefficient clusters—those spanning non-contiguous layers—drops from 8 in the baseline to 4 in our method (Table 1). The cluster size distribution (Figure 3) also shifts toward larger clusters, and the largest cluster size increases from 94 in the baseline to 179 in our method (Table 1), suggesting that our method supports competitive exploration around anchor features.

**Model Interpretability.** Grad-CAM++ visualizations (Figure 3) show that our method produces sharper and more class-focused attribution maps, whereas the baseline often highlights irrelevant background regions. Quantitatively, in the Point Game metric (Table 1), our method achieves 70.28% compared to the baseline's 61.25%, and in the Energy PG metric, 64.35% compared to 48.70%.

**Downstream Task Performance.** For CIFAR-10 classification accuracy (Table 1), our method reaches 97.58%, showing a slight improvement over the baseline's 97.49%. On ImageNet-1K (Table 1), our method attains a top-1 accuracy of 65.12% under the large-model short-epoch setting, again slightly exceeding the baseline's 65.00%.

Across all experiments, our method produced better-structured hidden features. The visualizations of features and clusters, together with per-cluster analysis of inefficient clusters, shows that similar features are predominantly explored within the same layer and propagate efficiently via identity mappings through residual connections. Per-layer cluster counts and per-cluster size distributions indicate that features are explored more tightly around representative anchors, suggesting that exploration is concentrated within the anchor's layer. Furthermore, per-layer difference analysis reveals that only representative features from earlier layers are retained and propagated, allowing deeper layers to focus on learning novel and complementary representations.

These behaviors yield clearer and more interpretable decision pathways, as corroborated by our model interpretability analysis. In addition, our method delivers slight improvements in downstream task performance, demonstrating that the enhanced interpretability is achieved without compromising accuracy.

## 5 CONCLUSION

We have presented a self-supervised learning framework that structures hidden feature representations by clustering rank-transformed activation patterns at the unit level and introducing a structure-aware regularization objective. By introducing representative anchor units and promoting their reuse across layers via residual connections, our method encourages similar features to be predominantly explored within the same layer and to propagate efficiently across layers. This architecture-agnostic approach enhances interpretability without compromising accuracy.

The main limitation of our work is the modest performance improvement, despite the interpretability gains achieved. To address this limitation, we aim to develop more effective structuring strategies. Specifically, we are investigating structuring strategies inspired by neuroscience and organization theory, analyzing how structures evolve under different training processes or changes in learning hyperparameters to identify actionable strategies, and further seeking to establish theoretically grounded structuring strategies based on information-theoretic principles.

In summary, our framework demonstrates that hidden feature representations can be effectively organized to enhance interpretability. However, the performance gains remain modest. To address this limitation, we plan to develop more effective structuring strategies. We hope that such efforts will establish structuring as a standard component in deep learning training. Moreover, this line of work may naturally extend to applications based on feature interpretation, such as controlling inference characteristics and pruning models to construct lightweight architectures.

**Ethics Statement.** This research does not involve human subjects, personally identifiable information, or sensitive personal data. Our datasets include a custom-designed synthetic dataset, CIFAR-10, and ImageNet-1K. The synthetic dataset is generated through fully reproducible procedures, and CIFAR-10 and ImageNet-1K are publicly available and widely used in the machine learning research community. Dataset usage complies with their respective licenses and terms of use. The proposed framework is architecture-agnostic and does not embed explicit or implicit demographic attributes, thereby minimizing risks of discrimination, bias, or fairness concerns. While improvements in deep learning could potentially be applied in sensitive domains, the current work is evaluated solely in image classification contexts for research purposes and does not address such applications. This research complies with the ICLR Code of Ethics.

**Reproducibility Statement.** We have taken extensive steps to ensure reproducibility. Detailed descriptions of the datasets, model architectures, hyperparameters, and training procedures are provided in Section 4 and Appendix B. All algorithmic components—sampling strategies, rank transformation, clustering, and the structure loss—are formally defined in Section 3 with accompanying pseudocode (Algorithm 1). To further facilitate verification and adoption, an anonymized, minimal version of the source code is shared as a zip archive for early access, and the full code will be released with the camera-ready version, together with a packaged library distribution designed for easy installation and broad applicability in downstream research.

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

## A    THE USE OF LARGE LANGUAGE MODELS

We used Large Language Models (LLMs) solely to improve the clarity and readability of this paper. Specifically, LLMs were applied for the refinement of grammar and terminology, and for identifying passages that might require additional clarification. LLMs were not used to generate research ideas, design methods, or produce original content.

## B    PREREQUISITE

### B.1    DATASETS

We evaluate our proposed framework using three datasets of varying complexity and scale: a synthetic dataset, CIFAR-10, and ImageNet-1K.

**Synthetic Dataset.** Inspired by the TensorFlow Playground (Hoeiness et al., 2021), we design a *spiral function* and additionally construct a *grid-based test set* for visualization evaluation. The spiral dataset enables non-linear binary classification in a low-dimensional setting and facilitates intuitive 2D visualization in Cartesian coordinates. Each class lies on a distinct branch of a two-dimensional spiral, offset by a phase shift of $\pi/2$, resulting in two interleaved spirals with a non-linear decision boundary (Figure 4). We generate $16{,}384$ training and $4{,}096$ test samples, equally split between the two classes. We deliberately keep the dataset size relatively small—much smaller than CIFAR-10 (60,000 samples)—to reflect the simplicity of the task.

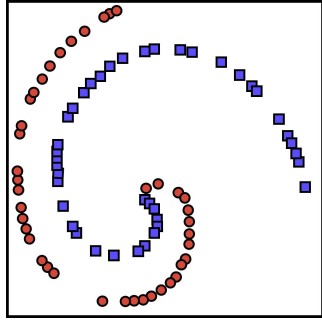

Figure 4: Spiral synthetic dataset. Visualization of 40 sampled data points per class from the synthetic dataset generated using a 2D spiral function. Red circles and blue squares represent the two class labels. This dataset provides a non-linear decision boundary for evaluating model representation structure.

Formally, data points are first generated in polar coordinates $(r, \phi)$ and then converted to Cartesian coordinates. For class 1, the angle is sampled as $\phi_1 \sim \mathcal{U}(0, 2\pi)$, while for class 2 the angle is phase-shifted by $\pi/2$, i.e., $\phi_2 = \phi_1 + \pi/2$. The radius is defined as $r = \phi_1/(2\pi)$, ensuring linear growth with respect to the angle. Finally, the polar coordinates are mapped to Cartesian coordinates:

$$\text{Class 1:} \quad (r \cdot \sin(\phi_1),\ r \cdot \cos(\phi_1))$$
$$\text{Class 2:} \quad (r \cdot \sin(\phi_2),\ r \cdot \cos(\phi_2))$$

We additionally construct a grid-based test set by uniformly sampling 50 equally spaced values along each axis in the range $[-1.0, 1.0]$, yielding 2,500 points. This dense grid allows for fine-grained visualization of model behavior both within and beyond the training manifold.

**CIFAR-10.** We use CIFAR-10 (Krizhevsky & Hinton, 2009), which consists of $60{,}000$ color images across 10 categories, with $50{,}000$ images for training and $10{,}000$ for testing. Each image has a resolution of $32 \times 32$ pixels. CIFAR-10 serves as a widely used benchmark for assessing both classification accuracy and interpretability in a real-world setting with manageable computational cost.

**ImageNet-1K.** As a standard-scale benchmark, ImageNet-1K (Russakovsky et al., 2015) provides about 1.28 million training images and 50,000 validation images, spanning 1,000 object categories. Images have varying resolutions, and during preprocessing, they are typically resized and cropped to $224 \times 224$ pixels. ImageNet-1K serves as a large-scale benchmark to evaluate the scalability and generalization of our framework to high-resolution, diverse, and complex real-world visual recognition tasks.

### B.2    HYPERPARAMETERS

We organize hyperparameters into four categories: model-related, dataset-related, training-related, and method-related. Model-related settings follow the `vit_wee` configuration from the PyTorch Image Models (timm) library by default, while dataset-related and training-related settings follow the `resnet50d` configuration from the same library. Few modifications are applied for the synthetic dataset and ImageNet-1K experiments to accommodate dataset characteristics and experimental constraints.

All experiments were conducted using Python 3.10.18 and PyTorch 2.7.1 with CUDA 12.8 on Ubuntu Linux. We used three computing systems: (1) NVIDIA RTX 3090 GPU, AMD EPYC 7502 32-core processor, and 377 GiB RAM; (2) NVIDIA RTX A6000 GPU, AMD EPYC 7513 32-core processor, and 1.0 TiB RAM; (3) NVIDIA A100 40GB GPU, AMD EPYC 7H12 64-core processor, and 1007 GiB RAM.

**Model-Related Hyperparameters.** By default, the token embedding dimension is set to 256, and the model consists of 14 transformer layers with 4 attention heads each. Dropout and drop path rates are set to 0.3 and 0.1, respectively, across all settings. To better accommodate the low-dimensional nature of the synthetic dataset, we reduce the patch size from the default $16 \times 16$ (used for image datasets) to $1 \times 1$, as each input is a single 2D point and does not require spatial decomposition. For ImageNet-1K, we use the `vit_mediumd` configuration from timm, with an embedding dimension of 512, 8 attention heads, and a transformer depth of 20, while keeping the patch size at $16 \times 16$. A summary of the model-related hyperparameters is provided in Table 2.

Table 2: Model-related hyperparameters.

| Hyperparameter | Synthetic Dataset | Image Dataset | Image Dataset (ImageNet-1K) |
|---|---|---|---|
| patch_size | $1 \times 1$ | $16 \times 16$ | $16 \times 16$ |
| embedding_dimension | 256 | 256 | 512 |
| transformer_depth | 14 | 14 | 20 |
| attention_heads | 4 | 4 | 8 |
| mlp_width_mult | 5 | 5 | 4 |
| dropout_rate | 0.3 | 0.3 | 0.3 |
| drop_path_rate | 0.1 | 0.1 | 0.1 |

**Dataset-Related Hyperparameters.** Both the synthetic and image datasets are normalized with a mean and standard deviation of 0.5, and smoothing is applied in both cases. The remaining preprocessing steps differ by dataset type.

For the image dataset, we follow the full data augmentation pipeline defined as the `rand_m8_inc1_mstd1.0` auto-augmentation policy, which selects one augmentation operation per image based on RandAugment. The base magnitude is 8, and slight variations are introduced by adding Gaussian noise with standard deviation 1.0 (via the `mstd` parameter). The `inc=1` flag restricts selection to augmentations whose severity increases meaningfully with magnitude (e.g., rotation, shear, brightness). In addition, we apply random scaling within $[0.08, 1.0]$, aspect ratio variation in $[0.75, 1.33]$, horizontal flipping with 50% probability, and small probabilities of grayscale conversion and Gaussian blur. Validation images are center-cropped with a crop ratio of 0.95.

For the synthetic dataset, which is low-dimensional, we disable geometric and color-based augmentations such as flipping, scaling, and blurring. Only normalization and smoothing are applied. Validation samples are not cropped. The complete set of dataset-related hyperparameters is provided in Table 3.

Table 3: Dataset-related hyperparameters.

| Hyperparameter | Synthetic Dataset | Image Dataset |
|---|---|---|
| mean | 0.5 | 0.5 |
| std | 0.5 | 0.5 |
| auto_augment | None | rand_m8_inc1_mstd1.0 |
| scale | [1.0, 1.0] | [0.08, 1.0] |
| ratio | [1.0, 1.0] | [0.75, 1.33] |
| hflip | 0 | 0.5 |
| grayscale_probability | 0 | 0.05 |
| gaussian_blur_probability | 0 | 0.05 |
| smoothing | 0.1 | 0.1 |
| crop_percentage (val) | 1.0 | 0.95 |

**Training-Related Hyperparameters.** All models are trained using mixed-precision training (AMP) with `float16` and the native backend. We use a batch size of 784 and the AdamW optimizer with `opt_betas=(0.6, 0.995)`, momentum 0.9, and a weight decay of 0.125. The main loss is defined as the cross-entropy classification loss. The random seed is fixed to 42 across all experiments, except for ImageNet-1K runs, where seeds $\{0, 1, 2\}$ are used.

The learning rate follows a cosine decay schedule (`sched=cosine`) with per-update adjustments (`sched_on_updates=true`). It is linearly increased from 0.0 during a separate 5-epoch warm-up phase, after which the base learning rate is set to 0.002. This value is scaled proportionally to the actual batch size (784) using a reference batch size of 4,096 (`lr_base_size=4096`), resulting in an effective initial learning rate of approximately 0.00038. For ImageNet-1K, the batch size is reduced to 392, resulting in an effective initial learning rate of approximately 0.00019 after scaling, which is about half of that in the other experiments.

By default, training is conducted for 3,600 epochs on both the synthetic and image datasets, preceded by a separate 5-epoch warm-up phase (`warmup_prefix=true`). For ImageNet-1K, we exceptionally define a reduced training configuration comprising 10 main epochs plus the same 5-epoch warm-up phase (15 epochs in total). Experiments on ImageNet-1K are run with multiple random seeds $[0, 1, 2]$, and the reported results are averaged over these seeds. Full details are provided in Table 4.

Table 4: Training-related hyperparameters.

| Hyperparameter | Synthetic Dataset | Image Dataset | Image Dataset (ImageNet-1K) |
|---|---|---|---|
| amp | true | true | true |
| amp_dtype | float16 | float16 | float16 |
| amp_impl | native | native | native |
| batch_size | 784 | 784 | 392 |
| classification_loss | cross-entropy | cross-entropy | cross-entropy |
| epochs | 3,600 | 3,600 | 10 |
| optimizer | adamw | adamw | adamw |
| opt_betas | [0.6, 0.995] | [0.6, 0.995] | [0.6, 0.995] |
| momentum | 0.9 | 0.9 | 0.9 |
| weight_decay | 0.125 | 0.125 | 0.125 |
| sched | cosine | cosine | cosine |
| sched_on_updates | true | true | true |
| seed | 42 | 42 | [0,1,2] |
| lr_base | 0.002 | 0.002 | 0.002 |
| lr_base_size | 4,096 | 4,096 | 4,096 |
| warmup_prefix | true | true | true |
| warmup_epochs | 5 | 5 | 5 |
| warmup_lr | 0.0 | 0.0 | 0.0 |

**Method-Related Hyperparameters.** We adopt hyperparameters to control the clustering of hidden features as listed in Table 5 for the synthetic datasets and Table 6 for the image datasets, where the naming conventions used in our code are mapped to their corresponding names in the main paper for clarity. By default, features are accumulated over 20 iterations (`n_accum_cluster = 20`) with a batch size of 784, resulting in $20 \times 784 = 15{,}680$ samples for clustering. This setting was chosen to approximate the size of the synthetic dataset; the algorithm was validated on this basis, and the same configuration is applied to other datasets for consistency. For ImageNet-1K, the training batch size is halved (392), yielding 7,840 clustering samples. When the required total exceeds the available dataset size, we instead use all samples (e.g., 4,096 for the synthetic test dataset; 2,500 for the grid-based test set).

To control computational cost for our method, we subsample tokens and layers, each with dimensionality 3 (`token_sampling_dim=3`, `layer_sampling_dim=3`). This subsampling captures local feature interactions within selected tokens and layers, while periodic random sampling ensures broader structural coverage throughout the model. During evaluation on the synthetic dataset, we subsample 14 layers (`layer_sampling_dim=14`) to capture structural patterns across all layers.

After feature collection, we perform rank-based comparisons by grouping feature values into a fixed number of bins (`n_bin=100`). The choice of 100 bins was determined by approximating the square root of the total number of samples for clustering in the synthetic training dataset (15,680) and rounding to the nearest hundred, which we empirically verified to yield stable and meaningful results; hence, the same setting is consistently adopted across all tasks. Features are grouped together if the sum of absolute rank differences across all samples is smaller than the threshold, which is defined as a proportion of the number of samples used for clustering (`scaling_threshold=1`). This effectively requires no more than an average rank difference of 1 per sample, resulting in a threshold $\tau$ of 15,680 in the default setting. The same criterion applies to other totals, such as 4,096 for the synthetic test dataset, 2,500 for the grid-based test set, and 7,840 for ImageNet-1K. Finally, we set the loss ratio $\gamma = 0.1$, which controls the relative contribution of the structure loss to the total training objective. This maintains a balance between structural constraints and the primary loss term.

Table 5: Algorithm-related hyperparameters for the synthetic datasets.

| Parameters (Name in Main Paper) | Synthetic Dataset Train | Synthetic Dataset Test | Synthetic Dataset Grid-Based Test |
|---|---|---|---|
| batch_size in Table 4 (train batch size $B_{train}$) | 784 | 784 | 784 |
| - (clustering batch size $B_{cluster}$) | 15,680 | 4,096 | 2,500 |
| n_accum_cluster ($B_{cluster} / B_{train}$) | 20 | 4,096/784 | 2,500/784 |
| layer_sampling_dim ($S_{\text{layer}}$) | 3 | 14 | 14 |
| token_sampling_dim ($S_{\text{token}}$) | 3 | 3 | 3 |
| n_bin (number of bins $\beta$) | 100 | 100 | 100 |
| scaling_threshold ($\tau/B_{cluster}$) | 1 | 1 | 1 |
| - (loss ratio $\gamma$) | 0.1 | 0.1 | 0.1 |

Table 6: Algorithm-related hyperparameters for the image datasets.

| Parameters | Image Dataset | Image Dataset (ImageNet-1K) |
|---|---|---|
| $B_{train}$ | 784 | 392 |
| $B_{cluster}$ | 15,680 | 7,840 |
| $B_{cluster} / B_{train}$ | 20 | 20 |
| $S_{\text{layer}}$ | 3 | 3 |
| $S_{\text{token}}$ | 3 | 3 |
| $\beta$ | 100 | 100 |
| $\tau/B_{cluster}$ | 1 | 1 |
| $\gamma$ | 0.1 | 0.1 |

## C  RESULTS

To rigorously assess the effectiveness of our proposed method, we conducted a series of statistical analyses. First, distribution-level tests (Kolmogorov–Smirnov and Mann–Whitney U) confirmed significant shifts in cluster size distributions, indicating more structured feature organization and stronger intra-layer competition. Second, layer-wise unit-level comparisons using both paired $t$-tests and permutation tests revealed statistically significant differences emerging from the mid to deep transformer layers, supporting our claim of layer-specific representational reshaping. Finally, task-level evaluations on Point Game and Energy PG demonstrated consistent and significant performance improvements, which also align with enhancements in model interpretability and the explanatory power of learned representations.

**Cluster Size Distribution.** To quantitatively support the shift in cluster size distribution observed in Figure 3 of the main paper, we conducted statistical comparisons between the baseline ViT and our method. The Kolmogorov–Smirnov test indicates a significant difference in the overall distribution of cluster sizes ($p = 1.673 \times 10^{-16}$). Additionally, the Mann–Whitney U test shows that our method yields significantly larger clusters on average ($p = 2.567 \times 10^{-22}$), supporting the claim that it promotes more structured feature organization and stronger intra-layer competition. The detailed statistics are summarized in Table 7.

Table 7: Statistical comparison of cluster size distributions between the baseline and our method. The total number of clusters is 8082 for the baseline ViT and 6478 for our method.

| Test | Statistic | p-value | Interpretation |
|---|---|---|---|
| Kolmogorov–Smirnov | 0.072 | $1.673 \times 10^{-16}$ | significant distribution difference |
| Mann–Whitney U | $2.398 \times 10^{7}$ | $2.567 \times 10^{-22}$ | ours significantly higher |

**Per-layer Difference.** To further support the layer-wise feature difference analysis presented in Table 1 of the main paper, we conducted statistical testing to assess the significance of unit-level differences between the baseline ViT and our proposed method. For each transformer layer, we applied two types of statistical tests.

First, we performed a paired $t$-test across the full set of unit differences per layer. This yielded extremely small $p$-values (typically smaller than $1 \times 10^{-300}$), which reflects the sensitivity of the $t$-test under large-$n$ conditions.

To complement this, and to provide a distribution-free validation, we also conducted a permutation test on a subsample of 5,000 unit pairs per layer, repeating the test 10,000 times. This approach evaluates the significance of observed mean differences without relying on parametric assumptions, and thus serves as a robustness check.

The resulting statistics are summarized in Table 8. The permutation-based $p$-values indicate statistically significant differences in unit-level representations beginning at layer 5 and continuing through layer 14 ($p < 0.05$), suggesting that the effect of our method becomes prominent in the deeper layers of the transformer.

These results provide further empirical evidence that the proposed method reshapes the internal feature structures of the transformer in a significant and layer-specific manner.

Table 8: Statistical comparison of layer-wise feature differences between the baseline and our method across transformer layers. For each layer, 3 sampled tokens were used, and each token's embedding consists of 256 dimensions, resulting in 768 unit-wise comparisons per layer. When comparing each of the 768 units against all 14 layers, this yields a total of 8,257,536 comparisons ($768 \times 14 \times 768$). Results are derived from full-set paired $t$-tests and subsampled permutation tests (5,000 samples, 10,000 iterations). Mean Difference (Mean Diff.) and Permutation Mean Difference (Perm. Mean Diff.) are computed as ViT – Ours.

| Layer | ViT | Ours | Diff | Perm. Diff | Perm. $p$-val | $t$-stat | $t$-test $p$-val |
|---|---|---|---|---|---|---|---|
| 1 | 166.97 | 162.32 | 4.65 | 3.48 | 0.0028 | 161.8752 | $< 1 \times 10^{-300}$ |
| 2 | 163.61 | 161.83 | 1.78 | 1.36 | 0.2478 | 61.2900 | $< 1 \times 10^{-300}$ |
| 3 | 161.64 | 161.85 | -0.21 | -0.52 | 0.6746 | -7.0965 | $1.28 \times 10^{-12}$ |
| 4 | 160.66 | 161.89 | -1.23 | -1.14 | 0.3446 | -41.0935 | $< 1 \times 10^{-300}$ |
| 5 | 159.54 | 161.94 | -2.40 | -3.02 | 0.0142 | -78.9885 | $< 1 \times 10^{-300}$ |
| 6 | 157.51 | 162.17 | -4.66 | -5.78 | $< 1 \times 10^{-4}$ | -153.3043 | $< 1 \times 10^{-300}$ |
| 7 | 156.02 | 162.30 | -6.28 | -7.37 | $< 1 \times 10^{-4}$ | -206.0222 | $< 1 \times 10^{-300}$ |
| 8 | 155.44 | 162.42 | -6.98 | -7.79 | $< 1 \times 10^{-4}$ | -229.6553 | $< 1 \times 10^{-300}$ |
| 9 | 153.43 | 162.43 | -9.00 | -9.76 | $< 1 \times 10^{-4}$ | -295.3185 | $< 1 \times 10^{-300}$ |
| 10 | 151.12 | 162.58 | -11.46 | -12.25 | $< 1 \times 10^{-4}$ | -373.0352 | $< 1 \times 10^{-300}$ |
| 11 | 150.80 | 162.76 | -11.96 | -12.57 | $< 1 \times 10^{-4}$ | -393.7288 | $< 1 \times 10^{-300}$ |
| 12 | 150.45 | 163.59 | -13.14 | -13.49 | $< 1 \times 10^{-4}$ | -445.6517 | $< 1 \times 10^{-300}$ |
| 13 | 150.19 | 162.74 | -12.55 | -12.47 | $< 1 \times 10^{-4}$ | -425.4399 | $< 1 \times 10^{-300}$ |
| 14 | 148.95 | 162.85 | -13.90 | -13.77 | $< 1 \times 10^{-4}$ | -469.4471 | $< 1 \times 10^{-300}$ |

**Point Game and Energy PG.** To further support the Point Game and Energy PG analysis presented in Table 1 of the main paper, we conducted statistical testing to assess the significance of performance differences between the baseline and our method. For each metric, we report the mean and standard deviation across trials, along with paired $t$-test statistics and $p$-values.

As summarized in Table 9, our method achieved higher scores in both. In the Point Game, the mean score improved from 0.613 to 0.703 ($p < 0.001$). In the Energy PG, the improvement was from 0.487 to 0.644 ($p < 0.001$).

These results demonstrate that the proposed method yields consistent and statistically significant gains, reinforcing our claim that it enhances the interpretability and explanatory power of the learned representations.

Table 9: Statistical comparison of Point Game and Energy PG between the baseline and our method. Mean $\pm$ standard deviation are reported, with paired $t$-test statistics and $p$-values shown separately.

| Metric | Base | Ours | $t$-stat | $p$-val |
|---|---|---|---|---|
| Point Game | $0.613 \pm 0.487$ | $0.703 \pm 0.457$ | $-13.517$ | $1.88 \times 10^{-41}$ |
| Energy PG | $0.487 \pm 0.344$ | $0.644 \pm 0.236$ | $-37.504$ | $< 1 \times 10^{-295}$ |

# D AUXILIARY ANALYSES

To support a more comprehensive understanding of our work, we provide auxiliary analyses organized into the following four sections. Section D.1 examines feature dynamics across layers using TCAV (Kim et al., 2018). Section D.2 extends our structure analysis to DINOv3 (Siméoni et al., 2025), illustrating that the proposed approach offers meaningful insights across diverse architectures and tasks. Section D.3 presents an ablation study on the feature transformation schemes used in our structure analysis, along with sensitivity analyses of the hyperparameters included in our method. Finally, Section D.4 provides supplementary explanations for Figure 1 to further clarify key aspects of the main results.

## D.1 FEATURE DYNAMICS ANALYSIS

To evaluate whether our method encourages the model to selectively rely on low-level concepts at the specific layer containing the concept-related anchor unit, we conduct an analysis using Testing with Concept Activation Vectors (TCAV) (Kim et al., 2018).

This analysis builds on prior findings showing that lower layers primarily learn low-level features (Zeiler & Fergus, 2014; Dosovitskiy et al., 2021). Accordingly, we adopt the Describable Textures Dataset (DTD) (Cimpoi et al., 2014), whose 47 texture categories closely correspond to such low-level representations, as our concept dataset.

For TCAV, each DTD class is treated as a distinct concept, and we compute layerwise TCAV scores for every CIFAR-10 class. For each target CIFAR-10 class, all remaining classes serve as non-concept examples. We sample 30 non-concept examples for each TCAV computation and repeatedly resample them 300 times without replacement to obtain 300 distinct combinations for each concept-class pair. This produces TCAV scores across 14 layers x 47 concepts $\times$ 10 target classes $\times$ 300 combinations. We then perform a layerwise paired t-test over all TCAV scores ($47 \times 10 \times 300$), and compute the win rate over concept-class pairs ($47 \times 10$) for which our method achieves a higher mean TCAV score.

As shown in Table 10, our model demonstrates significantly higher TCAV scores at layer 5, indicating more focused utilization of low-level concepts. These findings quantitatively support our claim that, through the use of anchors, features become more concentrated and effectively learned at specific layers.

Table 10: TCAV-based analysis of layerwise utilization of low-level concepts. We report mean TCAV scores for ViT and our method on CIFAR-10, along with their differences, paired $t$-statistics, $p$-values, and win rates (percentage). The layer exhibiting the largest performance gap is highlighted in **bold**.

| Layer | ViT | Ours | Diff | $t$-stat | $p$-val | Win |
|---|---|---|---|---|---|---|
| 1 | 0.49406 | 0.49327 | -0.00080 | -1.59 | 1.13e-01 | 44.26 |
| 2 | 0.49313 | 0.49306 | -0.00007 | -0.13 | 8.96e-01 | 48.72 |
| 3 | 0.49206 | 0.49392 | 0.00186 | 3.46 | 5.49e-04 | 55.96 |
| 4 | 0.48993 | 0.49350 | 0.00357 | 6.19 | 6.12e-10 | 61.70 |
| **5** | **0.48761** | **0.49392** | **0.00631** | **10.51** | **7.99e-26** | **68.94** |
| 6 | 0.48998 | 0.49285 | 0.00286 | 4.63 | 3.67e-06 | 59.15 |
| 7 | 0.48852 | 0.48805 | -0.00048 | -0.72 | 4.69e-01 | 48.09 |
| 8 | 0.48869 | 0.48772 | -0.00097 | -1.47 | 1.41e-01 | 46.17 |
| 9 | 0.48905 | 0.48597 | -0.00308 | -4.67 | 3.05e-06 | 41.70 |
| 10 | 0.49083 | 0.48811 | -0.00273 | -4.28 | 1.89e-05 | 44.04 |
| 11 | 0.49402 | 0.49153 | -0.00249 | -4.11 | 3.97e-05 | 44.89 |
| 12 | 0.50087 | 0.49821 | -0.00266 | -4.37 | 1.27e-05 | 42.34 |
| 13 | 0.51437 | 0.51423 | -0.00014 | -0.16 | 8.75e-01 | 48.72 |
| 14 | 0.54147 | 0.54442 | 0.00294 | 1.88 | 5.94e-02 | 54.89 |

## D.2 STRUCTURE COMPARISON WITH DINOV2 `DINO.TXT` AND DINOV3 `DINO.TXT`

To verify that our structure analysis generalizes across different architectures and tasks, we compare DINOv2 `dino.txt` (Jose et al., 2025) and DINOv3 `dino.txt` (Siméoni et al., 2025), which share the same backbone architecture but differ in downstream performance, training methodology, and the inclusion of an additional text encoder trained for text-related tasks.

We conduct our structure analysis on the ImageNet-1k validation set. Since the DINO `dino.txt` models are substantially larger than the `vit_wee` architecture used in our main experiments, we include all transformer blocks in the analysis but subsample features for tractability: for each layer, we use features from the first three patches and 2-dimensional slices from the embedding space, and employ a batch size of 4000 for structure computation.

As shown in Table 11, DINOv3 `dino.txt` achieves a 0.34% higher downstream performance than DINOv2 `dino.txt` despite having the same model size. Table 12 further shows that both the vision backbone and the text encoder (trained on top of the frozen vision backbone) exhibit larger cluster-size distributions in DINOv3 `dino.txt`, with average differences of 0.007 and 0.107, respectively. Although the absolute differences in mean cluster size appear small due to the dominance of clusters of size 1, the histogram in Figure 5 reveals a clear shift toward larger clusters. This trend is consistent with our main finding that larger cluster sizes correspond to competitive exploration among units related to a specific feature and are associated with improved task performance.

Table 11: Top-1 classification accuracy (percentages) on the ImageNet-1k validation set for DINOv2 `dino.txt` and DINOv3 `dino.txt`. The values in the table are our reproduced results, while the original papers report 81.6% for DINOv2 `dino.txt` and 82.3% for DINOv3 `dino.txt`. **Bold** indicates the better result.

|      | DINOv2 `dino.txt` | DINOv3 `dino.txt` |
| ---- | ----------------- | ----------------- |
| Acc. | 81.59             | **81.93**         |

Table 12: Mean size of the cluster. **Bold** indicates the better result.

|                  | DINOv2 dino.txt | DINOv3 dino.txt |
| ---------------- | --------------- | --------------- |
| Vision backbone  | 1.009           | **1.016**       |
| Text encoder     | 3.005           | **3.112**       |

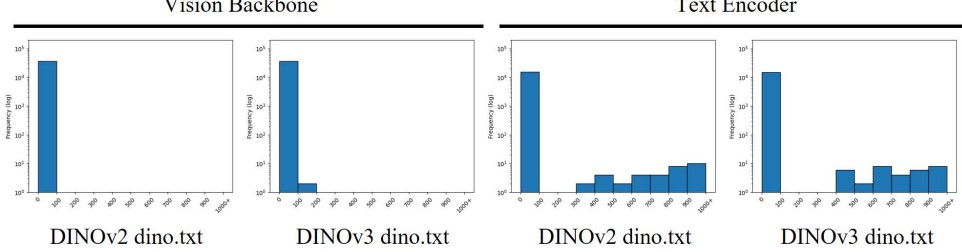

Figure 5: Cluster size distributions.

## D.3 Ablation and Hyperparameter Sensitivity

To more precisely quantify the contribution of each component in our algorithm, we conduct ablation studies and hyperparameter sensitivity experiments. All experiments are performed under a lighter setting than in the main paper: we train on CIFAR-10 for 300 epochs using a reduced batch size of 392.

Our ablation focuses on the grouped rank transformation introduced in Section 3.3. We evaluate variants in which this transformation is removed or modified to assess how these changes influence classification accuracy.

As shown in Table 13, only Ours_392—our method equipped with the grouped rank transformation and appropriate hyperparameters—achieves an improved accuracy over ViT (a gain of 0.59%). All other variants result in performance degradation. This results indicate that the grouped rank transformation is essential for stable structure estimation, as it mitigates sensitivity to absolute feature magnitudes and feature-level noise, consistent with the discussion in the main text.

Table 13: Ablation study on the feature transformation schemes used in our structure analysis. We report Top-1 CIFAR-10 classification accuracy (percentages) for ViT and several variants: rank applies standard ranking without grouping, raw uses unprocessed feature values, and normalized raw applies batch-wise normalization to raw features. Ours_392 and Ours_15680 differ only in the cluster batch size used during structure computation (see Table 14). **Bold** indicates the best-performing method.

|      | ViT   | Ours_392  | Ours_15680 | Rank  | Raw   | Normalized raw |
|------|-------|-----------|------------|-------|-------|----------------|
| Acc. | 92.43 | **93.02** | 91.69      | 91.60 | 87.31 | 92.14          |

To analyze hyperparameter sensitivity, we conduct experiments on two key factors in our structure analysis: the batch size used when estimating structure, and the threshold on the average rank difference used to determine whether two units belong to the same cluster.

As shown in Table 14, using a structure-analysis batch size that matches the training batch size (392) yields the best performance. While the main experiments in the paper reuse the hyperparameter settings that were validated on the synthetic dataset, these results suggest that additional performance gains may be achievable through dedicated hyperparameter tuning. Moreover, Figure 6 demonstrates that, regardless of the hyperparameter choice, our method consistently improves more rapidly than the ViT baseline during the first 30 epochs. The best-performing hyperparameter setting (batch size 392) also provides more stable and higher accuracy throughout training.

Table 14: Top-1 CIFAR-10 classification accuracy (percentages) under different batch sizes used for structure analysis. Ours_392, Ours_7840, and Ours_15680 correspond to using 392, 7,840, and 15,680 samples, respectively, when computing the structure. **Bold** indicates the best-performing hyperparameter setting.

| Clustering batch size ($B_{cluster}$) | Ours_392  | Ours_7840 | Ours_15680 |
|---------------------------------------|-----------|-----------|------------|
| Acc.                                  | **93.02** | 91.30     | 91.69      |

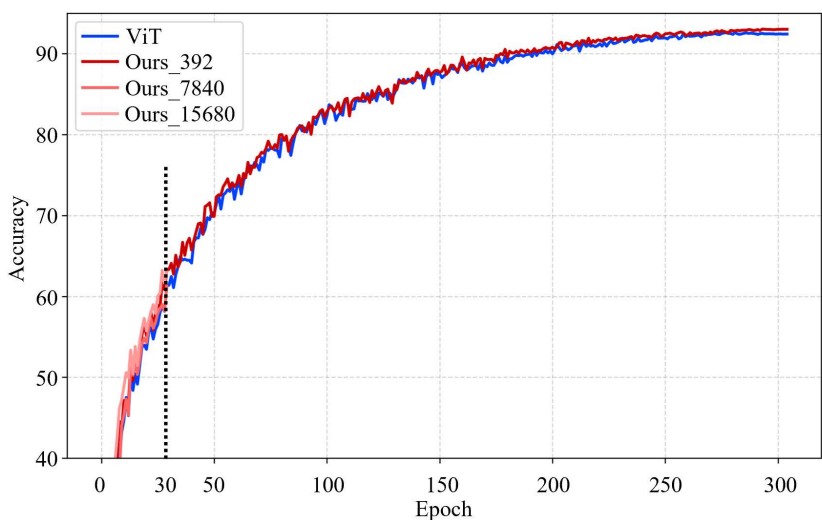

Figure 6: Accuracy curve.

To further assess the sensitivity of our method to the criterion used for merging units into a single cluster, we conduct an experiment in which the average-rank difference threshold is reduced by half. As shown in Table 15, retaining the original threshold value used in the main paper (i.e., 1) consistently yields superior performance.

Table 15: Top-1 CIFAR-10 classification accuracy (percentages) under different scaling thresholds. Each value corresponds to the difference threshold $\tau$ in Section 3.3: a threshold of $0.5$ corresponds to $15{,}680 \times 0.5 = 7{,}840$, whereas a threshold of $1.0$ corresponds to $15{,}680 \times 1.0 = 15{,}680$. **Bold** indicates the best-performing hyperparameter setting.

| Scaling threshold ($\tau/B_{cluster}$) | 0.5 | 1.0 |
|---|---|---|
| Acc. | 87.28 | **91.69** |

## D.4  ADDITIONAL VISUALIZATION

To supplement the discussion of the clusters marked with bold red Xs in Figure 1, we provide an additional example illustrating the ineffective feature reuse, characterized by an interspersed pattern in which units briefly return to similar feature distributions but are interrupted by layers exhibiting substantially different behavior. This example uses a different architecture composed of 32 linear layers with residual connections and a hidden dimension of 32.

As shown in Figure 7, the regions highlighted with red circles illustrate such ineffective reuse. For Unit A, Layers 3 and 5 display similar activation values; for Unit B, Layers 23 and 26 behave similarly; and for Unit C, Layers 25 and 30 also exhibit comparable values. However, the intermediate layers between these pairs (e.g., Layer 4 for Unit A) do not maintain this similarity and instead produce noticeably different activation values. This interspersed pattern contrasts with the effective feature reuse enabled by our method via identity mappings through residual connections, where related features propagate more coherently across layers.

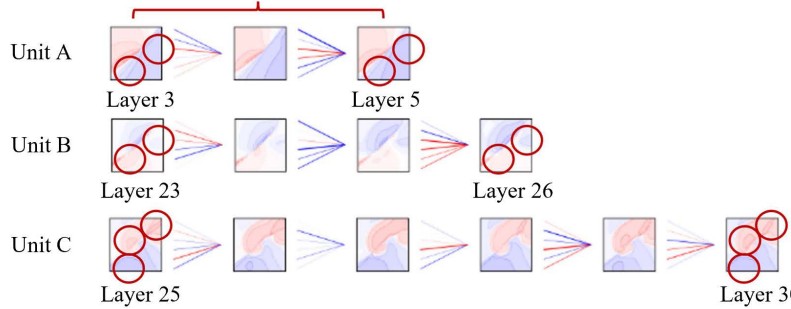

Figure 7: Visualization of hidden features in a model with 32 linear layers and residual connections, each with a hidden dimension of 32. Red circles highlight regions that primarily influence clustering in our structure analysis. For additional context, a subset of inter-layer weights is visualized, where line thickness represents magnitude and color denotes sign.

