# OpenReview forum: "Structuring Hidden Features via Clustering of Unit-Level Activation Patterns"
_ICLR.cc/2026/Conference — Submitted to ICLR 2026_

### Official Review · Reviewer_ZjDG · 2025-10-31

**Soundness:** 2
**Presentation:** 4
**Contribution:** 2
**Rating:** 4
**Confidence:** 4

**Summary:**

Deep neural networks develop complex and unstructured internal representations, often creating redundant features that are difficult to interpret. This paper introduces a self-supervised regularization method to better organize hidden features, enabling their reuse across layers and increasing feature diversity within layers. This approach improves interpretability, makes better use of network resources, and may enhance generalization performance.

The method has two main components: First, it identifies redundant features through cross-layer clustering. Second, it implements a structure-aware regularization that encourages the reuse of one unit per cluster through residual connections while allowing other units to learn complementary features.

The authors tested their approach on three datasets: a synthetic task, CIFAR-10, and ImageNet, using variants of the ViT architecture. They developed new metrics to measure feature reuse, diversity, while utilizing previously proposed metrics for interpretability, and performance. Compared to standard training methods, their results showed better feature organization and interpretability.

**Strengths:**

- The paper presents its ideas clearly and comprehensively, with excellent organization and complete details.
- The approach is novel, introducing efficient methods to reduce computational costs without compromising effectiveness. The use of group ranked transformation for clustering helps reduce sensitivity to magnitude differences. The evaluation framework and analysis metrics are well-designed.
- The concept of enabling precise unit-level feature reuse across layers while utilizing residual layers is particularly novel.

**Weaknesses:**

- The paper's primary weakness lies in its limited experimental scope. While the presented results are promising, a broader evaluation across diverse datasets, model architectures, and network layers would better demonstrate the method's generalizability and practical impact. Enhanced visualizations of feature organization across multiple layers would also strengthen the paper's empirical validation.

- The introduction of multiple hyper-parameters without detailed ablation studies makes it challenging to determine optimal settings for future applications.

**Questions:**

- Is the structure loss calculated at the sample level?
- When clustering flattened representations across token positions and layers, multiple units from different token positions can end up in the same cluster, but only one anchor unit is selected per cluster. Would selecting multiple anchor units per cluster for each unique token position improve results?
- How does the method handle cases where the same unit position from different layers appears in different clusters but has the lowest index in each? This could create multiple loss computations for the same residual stream position.
- How does this method perform with text inputs, where token counts vary and token positions can have significantly different representations? A discussion is required on the applicability across domains.

---

> ### Author Response · Authors · 2025-11-19
>
> Thank you for the reviewers’ efforts in evaluating our paper,  for recognizing the novelty of our approach, and for evaluating its presentation as having excellent organization. We have prepared the following responses to the Weakness and Questions raised in the review. We have also updated the paper by adding the relevant details as a new section in the appendix.
>
> > While the presented results are promising, a broader evaluation across diverse datasets, model architectures, and network layers would better demonstrate the method's generalizability and practical impact.
>
> To demonstrate that our method remains meaningful across diverse settings, we analyze the structures of *DINOv2 dino.txt* and *DINOv3 dino.txt*—models that share the same architecture but differ in interpretability and performance. Specifically, we examine whether *DINOv3*, which is known to exhibit stronger interpretability and higher accuracy, also shows more coherent internal structure consistent with our experimental trends. These results are provided in Appendix D.2.
>
> The findings indicate that models with more coherent structures (i.e., larger clusters) tend to achieve better performance:
>
> | Mean cluster size | DINOv2 dino.txt | DINOv3 dino.txt |
> | --- | --- | --- |
> | Vision backbone | 1.009 | **1.016** |
> | Text encoder | 3.005 | **3.112** |
>
> | Imagenet-1k | DINOv2 dino.txt | DINOv3 dino.txt |
> | --- | --- | --- |
> | Acc. | 81.59 | **81.93** |
>
> Taken together, these findings indicate that better interpretability and improved performance are closely linked to more coherent internal structure, reinforcing the relevance of our structural analysis.
>
> > Enhanced visualizations of feature organization across multiple layers would also strengthen the paper's empirical validation.
>
> For enhanced visualizations of feature organization, we added Figure 7 in Appendix D.4. To clarify what it means for features to be separated into different clusters and to illustrate cases of ineffective feature reuse, we highlight relevant regions with red circles to show how features change across layers. In addition to visualizing selected units, we also analyze layer-level feature dynamics using TCAV, which provides further insight into feature organization. These results are provided in Appendix D.1.
>
> The TCAV analysis shows that low-level features exhibit a substantially higher concentration at Layer 5 compared to the baseline ViT.
>
> | Layer (Block) | ViT | Ours | Diff | t-stat | p-val | win |
> | --- | --- | --- | --- | --- | --- | --- |
> | 1 | 0.49406 | 0.49327 | -0.00080 | -1.59 | 1.13e-01 | 44.26 |
> | 2 | 0.49313 | 0.49306 | -0.00007 | -0.13 | 8.96e-01 | 48.72 |
> | 3 | 0.49206 | 0.49392 | 0.00186 | 3.46 | 5.49e-04 | 55.96 |
> | 4 | 0.48993 | 0.49350 | 0.00357 | 6.19 | 6.12e-10 | 61.70 |
> | 5 | **0.48761** | **0.49392** | **0.00631** | **10.51** | **7.99e-26** | **68.94** |
> | 6 | 0.48998 | 0.49285 | 0.00286 | 4.63 | 3.67e-06 | 59.15 |
> | 7 | 0.48852 | 0.48805 | -0.00048 | -0.72 | 4.69e-01 | 48.09 |
> | 8 | 0.48869 | 0.48772 | -0.00097 | -1.47 | 1.41e-01 | 46.17 |
> | 9 | 0.48905 | 0.48597 | -0.00308 | -4.67 | 3.05e-06 | 41.70 |
> | 10 | 0.49083 | 0.48811 | -0.00273 | -4.28 | 1.89e-05 | 44.04 |
> | 11 | 0.49402 | 0.49153 | -0.00249 | -4.11 | 3.97e-05 | 44.89 |
> | 12 | 0.50087 | 0.49821 | -0.00266 | -4.37 | 1.27e-05 | 42.34 |
> | 13 | 0.51437 | 0.51423 | -0.00014 | -0.16 | 8.75e-01 | 48.72 |
> | 14 | 0.54147 | 0.54442 | 0.00294 | 1.88 | 5.94e-02 | 54.89 |
>
> These results support the claim that our method encourages the emergence of representative units that act as anchors for related features.
>
> > The introduction of multiple hyper-parameters without detailed ablation studies makes it challenging to determine optimal settings for future applications.
>
> We added experiment results on hyperparameter sensitivity in Appendix D.2.
>
> For the clustering batch size, the results are as follows. The best performance is obtained when the clustering batch size equals the training batch size. (In the main experiments, we generally used 15,680 samples, which produced meaningful results on the synthesized dataset by using nearly the entire training set.)
>
> | CIFAR-10 | Ours_392 | Ours_7840 | Ours_15680 |
> | --- | --- | --- | --- |
> | Acc. | **93.02** | 91.30 | 91.69 |
>
> For the scaling threshold, the results show that a threshold of 1.0 (i.e., an average difference of 1) yields the best performance, which is consistent with the discussion in the paper.
>
> | CIFAR-10 | 0.5 | 1.0 |
> | --- | --- | --- |
> | Acc. | 87.28 | **91.69** |
>
> Overall, these hyperparameter studies indicate that (i) further gains may be achieved through additional exploration of the clustering batch size, while (ii) the scaling threshold used in the paper already provides the most effective setting.

---

> > ### Author Response · Authors · 2025-11-19
> >
> > > Is the structure loss calculated at the sample level?
> >
> > Yes. The structure loss is computed at the sample level, as explained in Sections 3.4 and 3.5.
> >
> > > Would selecting multiple anchor units per cluster for each unique token position improve results?
> >
> > This is indeed possible. Exploring more complex structuring strategies beyond the one proposed in the paper could be a promising direction for future work.
> >
> > > How does the method handle cases where the same unit position from different layers appears in different clusters but has the lowest index in each? This could create multiple loss computations for the same residual stream position.
> >
> > In our setup, this issue does not occur because we analyze only three consecutive layers during training. More precisely, given layers A, B, and C, the possible cluster compositions are (A), (B), (C), (A, B), (B, C), (A, C), and (A, B, C). After obtaining the connected components, units with the same index and similar features are grouped into the same cluster; therefore, cases that would result in multiple loss computations for the same unit position are naturally excluded.
> >
> > However, if many more layers were included for structure analysis during training, such conflicts could arise. For example, even after obtaining connected components, clusters such as A–C and B–D could remain, leading to multiple losses between Layers B and C. In such cases, one would need either an algorithm to resolve multiple-loss computations or a mechanism to choose optimal anchors rather than simply selecting the lowest-index unit.
> >
> > > How does this method perform with text inputs, where token counts vary and token positions can have significantly different representations?
> >
> > In our response to the comment *“While the presented results are promising, a broader evaluation across diverse datasets, model architectures, and network layers would better demonstrate the method's generalizability and practical impact.”* we confirm that meaningful structural analysis can indeed be applied to models trained with text inputs.

---

### Official Review · Reviewer_cEKB · 2025-10-31

**Soundness:** 2
**Presentation:** 2
**Contribution:** 1
**Rating:** 2
**Confidence:** 4

**Summary:**

This paper proposes a self-supervised learning framework aimed at improving the interpretability of deep neural networks by structuring hidden feature representations. The method operates at the hidden-unit level, clustering activation patterns across data samples and imposing a structure-aware regularization that encourages cross-layer feature reuse and the emergence of representative anchor units.

**Strengths:**

1. Structured feature representation is an interesting topic. The paper does introduce a hidden-unit-level approach to organize features though it may remain complex.
2. The combination of clustering hidden units and enforcing structure via a regularization objective is conceptually interesting and aligns with efforts to improve interpretability through learned representations.

**Weaknesses:**

1. The evaluation relies heavily on Grad-CAM++ metrics. Gradient-based attribution methods are known to have limitations (especially in deep networks) and can produce misleading explanations. This raises concerns about whether the reported scores in interpretability are meaningful.

2. The paper does not adequately position itself relative to prior explanation methods. Many traditional explanation methods are missing, such as TCAV (Kim et al.) and DINO. While these methods are not specifically relevant to "structure", they are helpful for understanding features.

3. The paper does not convincingly demonstrate that structured representations are helpful for downstream tasks. If not, the advantage of structured features over existing feature characterization methods can be the key. But this part is missing in the current scope.

4. I am curious about the impact of the structure-aware regularization on feature dynamics and learning efficiency.

**Questions:**

See above

---

> ### Author Response · Authors · 2025-11-19
>
> Thank you for the reviewers’ efforts in evaluating our paper and for suggesting ways to more clearly highlight its significance. We have prepared the following responses to address the points raised in the Strength, Weakness, and Questions sections. We have also updated the paper by adding the relevant details as a new section in the appendix.
>
> > it may remain complex
>
> We acknowledge that the procedure for meaningfully analyzing structures and improving them may appear complex. However, each component corresponds to design choices that our experiments have shown to be necessary as in Appendix D.3.
>
> To address potential concerns about complexity and to facilitate future use or extension of our method, we maintain the latest version of the code as a library. After the anonymity period, we plan to release it on GitHub and register it on PyPI. The method can be applied easily, as illustrated below:
>
> ```
> from structure_features import create_model, create_loss
>
> # Wrap a base model
> structure_model = create_model(model)
>
> # Forward pass with features and structures
> y, features, structures = structure_model(x) # original code : y = model(x)
>
> # Compute structure loss
> structure_loss = create_loss()
> loss = structure_loss(features, structures)
> ```
>
> Furthermore, the codebase is modularized according to the subsections described in Section 3, and documentation is included to improve clarity.
>
> > The evaluation relies heavily on Grad-CAM++ metrics. Gradient-based attribution methods are known to have limitations (especially in deep networks) and can produce misleading explanations.
>
> To strengthen the validity of our claims beyond Grad-CAM++, we analyzed whether interpretability performance is meaningfully associated with our structural metrics. To do so, we leveraged the differences between DINOv2 and DINOv3, two models of the same size for which improved interpretability has been reported. Specifically, we analyzed the internal structures of  DINOv2 dino.txt and DINOv3 dino.txt. These results are provided in Appendix D.2.
>
> The results are as follows. We observe that the vision backbone of DINOv3 dino.txt exhibits a larger mean cluster size compared to DINOv2 dino.txt:
>
> | Mean cluster size | DINOv2 dino.txt | DINOv3 dino.txt |
> | --- | --- | --- |
> | Vision backbone | 1.009 | **1.016** |
>
> Since most cluster sizes are 1, a histogram provides a clearer comparison than the mean alone. For this reason, we additionally include the corresponding histogram in Figure 5.
>
> Taken together, these findings indicate that more coherent internal structure is closely linked to better interpretability.
>
> > Many traditional explanation methods are missing, such as TCAV (Kim et al.) and DINO.
>
> In our response to the comment *“The evaluation relies heavily on Grad-CAM++ metrics. Gradient-based attribution methods are known to have limitations (especially in deep networks) and can produce misleading explanations.”* we clarified the relationship between our structural analysis and the explanations observed in DINO. We then conducted additional analysis using TCAV to examine feature dynamics, as provided in Appendix D.1.
>
> In particular, we investigated whether our training method enables the model to better utilize lower-level concepts at specific layers. We observed that the model trained with our method leverages lower-level concepts more effectively, with a notably higher concentration at Layer 5.
>
> | Layer (Block) | ViT | Ours | Diff | t-stat | p-val | win |
> | --- | --- | --- | --- | --- | --- | --- |
> | 1 | 0.49406 | 0.49327 | -0.00080 | -1.59 | 1.13e-01 | 44.26 |
> | 2 | 0.49313 | 0.49306 | -0.00007 | -0.13 | 8.96e-01 | 48.72 |
> | 3 | 0.49206 | 0.49392 | 0.00186 | 3.46 | 5.49e-04 | 55.96 |
> | 4 | 0.48993 | 0.49350 | 0.00357 | 6.19 | 6.12e-10 | 61.70 |
> | 5 | **0.48761** | **0.49392** | **0.00631** | **10.51** | **7.99e-26** | **68.94** |
> | 6 | 0.48998 | 0.49285 | 0.00286 | 4.63 | 3.67e-06 | 59.15 |
> | 7 | 0.48852 | 0.48805 | -0.00048 | -0.72 | 4.69e-01 | 48.09 |
> | 8 | 0.48869 | 0.48772 | -0.00097 | -1.47 | 1.41e-01 | 46.17 |
> | 9 | 0.48905 | 0.48597 | -0.00308 | -4.67 | 3.05e-06 | 41.70 |
> | 10 | 0.49083 | 0.48811 | -0.00273 | -4.28 | 1.89e-05 | 44.04 |
> | 11 | 0.49402 | 0.49153 | -0.00249 | -4.11 | 3.97e-05 | 44.89 |
> | 12 | 0.50087 | 0.49821 | -0.00266 | -4.37 | 1.27e-05 | 42.34 |
> | 13 | 0.51437 | 0.51423 | -0.00014 | -0.16 | 8.75e-01 | 48.72 |
> | 14 | 0.54147 | 0.54442 | 0.00294 | 1.88 | 5.94e-02 | 54.89 |
>
> Taken together, these results support the claim that our method encourages the emergence of representative units that act as anchors for related features.

---

> > ### Author Response · Authors · 2025-11-19
> >
> > > The paper does not convincingly demonstrate that structured representations are helpful for downstream tasks.
> >
> > As pointed out in “*Do Deep Generative Models Know What They Don’t Know?* (Eric Nalisnick et al.)”, models trained solely on CIFAR-10 may not reliably generalize to downstream tasks. For this reason, and to provide stronger evidence, we additionally analyzed downstream behavior using DINO. These results are provided in Appendix D.2.
> >
> > We observe that the text encoder trained with the frozen DINOv3 vision backbone, which exhibits more coherent internal structure, shows a larger mean cluster size and achieves higher accuracy on ImageNet-1k:
> >
> > | Mean cluster size | DINOv2 dino.txt | DINOv3 dino.txt |
> > | --- | --- | --- |
> > | Vision backbone | 1.009 | **1.016** |
> > | Text encoder | 3.005 | **3.112** |
> >
> > | ImageNet-1k | DINOv2 dino.txt | DINOv3 dino.txt |
> > | --- | --- | --- |
> > | Acc. | 81.59 | **81.93** |
> >
> > Taken together, these findings indicate that more coherent internal structure helps models learn better representations for downstream tasks and contributes to improved performance.
> >
> > > If not, the advantage of structured features over existing feature characterization methods can be the key. But this part is missing in the current scope.
> >
> > In our response to the comment *“Many traditional explanation methods are missing, such as TCAV (Kim et al.) and DINO.”* we addressed this concern by providing an analysis of feature dynamics using TCAV.
> >
> > > I am curious about the impact of the structure-aware regularization on feature dynamics and learning efficiency.
> >
> > In our response to the comment *“Many traditional explanation methods are missing, such as TCAV (Kim et al.) and DINO.”* we provide an analysis of feature dynamics using TCAV.
> >
> > Regarding learning efficiency, we added accuracy curves during training to Appendix D.3 (Figure 6). Across various hyperparameter settings, the structure-aware models consistently demonstrate faster performance improvement than the ViT baseline during the first 30 epochs. Under the best hyperparameter configuration—where the number of samples used for structure analysis matches the training batch size—the model generally maintains superior accuracy throughout the entire training process.

---

### Official Review · Reviewer_uggD · 2025-11-01

**Soundness:** 2
**Presentation:** 2
**Contribution:** 2
**Rating:** 4
**Confidence:** 3

**Summary:**

This paper propose a novel way to align the feature across many different layers of a neural network. The author propose to collect a latent embedding buffer during training, which contains the embedding across different sample, position, and layers, then cluster these embedding to create "feature anchor" points. Then the author define an auxiliary loss to constraint the latent of the neural network to match these feature anchor accordingly. The author claims the this auxiliary loss makes the model more explainable in two way: 1. features across different layers are more aligned. 2. model trained auxiliary loss when applied with grad-cam produces better unsupervised segmentation map.

**Strengths:**

The author present a novel way to make neural network more interpretable.
Method presented by the author is not post-hoc, unlike many other interpretability works.
The author shows their method aligned with class-level segmentation map better, when applied grad-CAM.

**Weaknesses:**

The author only provide the baseline against VIT trained with standard classification loss, but did not compare their method with other method that improves model's interpretability.
The model does
The author does not provide standard evaluation (classification accuracy) between standard VIT training and model trained with their auxilary loss.
There are many moving part of the design, ranking as preprocessing, and group rank, the author did not provide enough ablation study to show these design are necessary.

**Questions:**

Can author read my summary to see if my understanding is correct? If not, please tell me and also explain to me how the model actually works.
I would guess the training with auxiliary will improve model's interpretability but hurt the model's performance, if so, how much?
The ranking part is confusing to me. Why do the author use "rank" to preprocess the latent embedding, then something like normalization?

---

> ### Author Response · Authors · 2025-11-19
>
> Thank you for the reviewers’ efforts in evaluating our paper and for recognizing the novelty of our work. To address the weaknesses and questions raised in the review, we provide the following clarifications. We have additionally updated the paper by adding the relevant details as a new section in the appendix.
>
> > The author only provide the baseline against VIT trained with standard classification loss, but did not compare their method with other method that improves model's interpretability.
>
> To address this concern, we apply our structure identification method for DINOv3, for which improvements in interpretability have been reported. Specifically, we include structural comparisons between DINOv2 dino.txt and DINOv3 dino.txt to examine whether models with higher interpretability also exhibit stronger internal structure. These results are provided in Appendix D.2.
>
> As a result, we observe that DINOv3 dino.txt exhibits larger mean cluster sizes for both the vision backbone and the text encoder:
>
> | Mean cluster size | DINOv2 dino.txt | DINOv3 dino.txt |
> | --- | --- | --- |
> | Vision backbone | 1.009 | **1.016** |
> | Text encoder | 3.005 | **3.112** |
>
> Taken together, these findings indicate that better interpretability is closely linked to more coherent internal structure, reinforcing the relevance of our structural analysis.
>
> > The model does The author does not provide standard evaluation (classification accuracy) between standard VIT training and model trained with their auxilary loss.
>
> This comparison is provided in Table 1. Applying our method yields improvements of +0.09% on CIFAR-10 and +0.12% on ImageNet-1K over standard ViT training.
>
> > There are many moving part of the design, ranking as preprocessing, and group rank, the author did not provide enough ablation study to show these design are necessary.
>
> To address this concern, we conduct ablation studies on different types of feature values used for identifying structures via unit-level clustering. These results are presented in Appendix D.3.
>
> Specifically, we compare rank, raw, and normalized raw representations. We observe that only our method (Ours_392), which sets the cluster batch size equal to the training batch size, outperforms the baseline ViT:
>
> | CIFAR-10 | ViT | Ours_392 | Ours_15680 | Rank | Raw | Normalized raw |
> | --- | --- | --- | --- | --- | --- | --- |
> | Acc. | 92.43 | **93.02** | 91.69 | 91.60 | 87.31 | 92.14 |
>
> These results support that our design choices are necessary for achieving performance gains.
>
> > Can author read my summary to see if my understanding is correct?
>
> We reviewed your summary and confirm that your understanding is accurate. To provide additional clarification, the part you summarized as “features across different layers are more aligned” can be more precisely interpreted—similar to an abstract—as indicating that our method (i) promotes feature reuse across layers via identity mappings, and (ii) encourages the emergence of representative units that serve as anchors for related features. The corresponding results can be found in Section 4.4.
>
> In addition, regarding point (ii), beyond showing that the differences between layers remain large, we further analyze how lower-level features are learned at specific layers using a TCAV-based approach. These results are provided in Appendix D.1.
>
> Specifically, using TCAV, we examine whether our method strengthens the use of lower-level concepts that are concentrated in specific layers. As a result, we find that low-level features exhibit a substantially higher concentration at layer 5 compared to the baseline ViT.
>
> | Layer (Block) | ViT | Ours | Diff | t-stat | p-val | win |
> | --- | --- | --- | --- | --- | --- | --- |
> | 1 | 0.49406 | 0.49327 | -0.00080 | -1.59 | 1.13e-01 | 44.26 |
> | 2 | 0.49313 | 0.49306 | -0.00007 | -0.13 | 8.96e-01 | 48.72 |
> | 3 | 0.49206 | 0.49392 | 0.00186 | 3.46 | 5.49e-04 | 55.96 |
> | 4 | 0.48993 | 0.49350 | 0.00357 | 6.19 | 6.12e-10 | 61.70 |
> | 5 | **0.48761** | **0.49392** | **0.00631** | **10.51** | **7.99e-26** | **68.94** |
> | 6 | 0.48998 | 0.49285 | 0.00286 | 4.63 | 3.67e-06 | 59.15 |
> | 7 | 0.48852 | 0.48805 | -0.00048 | -0.72 | 4.69e-01 | 48.09 |
> | 8 | 0.48869 | 0.48772 | -0.00097 | -1.47 | 1.41e-01 | 46.17 |
> | 9 | 0.48905 | 0.48597 | -0.00308 | -4.67 | 3.05e-06 | 41.70 |
> | 10 | 0.49083 | 0.48811 | -0.00273 | -4.28 | 1.89e-05 | 44.04 |
> | 11 | 0.49402 | 0.49153 | -0.00249 | -4.11 | 3.97e-05 | 44.89 |
> | 12 | 0.50087 | 0.49821 | -0.00266 | -4.37 | 1.27e-05 | 42.34 |
> | 13 | 0.51437 | 0.51423 | -0.00014 | -0.16 | 8.75e-01 | 48.72 |
> | 14 | 0.54147 | 0.54442 | 0.00294 | 1.88 | 5.94e-02 | 54.89 |
>
> Taken together, these results support the claim that our method encourages the emergence of representative units that act as anchors for related features.

---

> > ### Author Response · Authors · 2025-11-19
> >
> > > I would guess the training with auxiliary will improve model's interpretability but hurt the model's performance, if so, how much?
> >
> > In our responses to the comments *“The model does The author does not provide standard evaluation (classification accuracy) between standard VIT training and model trained with their auxilary loss. ”* and *“There are many moving part of the design, ranking as preprocessing, and group rank, the author did not provide enough ablation study to show these design are necessary.”* we have shown that our method not only preserves performance but indeed improves it.
> >
> > > Why do the author use "rank" to preprocess the latent embedding, then something like normalization?
> >
> > In our response to the comment, *“There are many moving part of the design, ranking as preprocessing, and group rank, the author did not provide enough ablation study to show these design are necessary.”* we presented ablation studies demonstrating the necessity of incorporating group rank into our method.

---

### Meta-Review · Area_Chair_3gbG · 2026-01-05

**Summary:**

**Summary** \
The paper proposes an unsupervised regularization mechanism aimed at improving the interpretability of the learnt representations.
The strategy first clusters the features from different samples, positions and layers to obtain anchor prototypes, and then regularise the learning of the represetations to match these anchors. This strategy enhances interpretability as measured by gradient-based visual explanations while retaining discriminative performance.

**Summary of Concerns** \
All reviewers appreciated the promising use of clustering and regularization to enhance the interpretability of neural network representations. However, they raised several important concerns that are not yet convincingly addressed:
1. **Quality / Completeness / Validity** The experimental scope is limited and insufficient to convincingly support the claims of improved interpretability (Reviewers uggD, cEKB, ZjDG). The authors provided additional analyses of DINOv2 and DINOv3 features, noting that mean cluster size correlates with improved classification performance. However, this analysis is reassuring but not conclusive: first, it does not establish statistical confidence in the observed correlation, and second, correlation does not imply causation. As suggested by Reviewer ZjDG, a more thorough evaluation across diverse datasets, model architectures, and network layers would be necessary to improve the validity and generalizability of the findings.
2. **Quality** Sensitivity analyses regarding hyperparameter choices and ablation studies remain underexplored and may be strongly model- and data-dependent. While the additional experiments provided in the rebuttal (Appendix D.2 and D.3) help complete the evaluation on ImageNet-like datasets and ViT architectures, they do not fully address the core issue of identifying the boundary conditions under which the proposed method is effective.

**Decision** \
The paper proposes an interesting, intuitive and yet promising solution to increase the interpretability of representations in deep learning. However, the findings from the paper are preliminary and the analysis needs to be expanded in scope in order to claim more general validity and identify their boundary conditions. At the moment, the quality of the experimental methodology does not meet the bar of acceptance to ICLR.

**Reviewer Concerns:**

As mentioned above, the main core issues have not been adequately addressed.

**Reviewer Scores:**

All reviewers would have kept their score, as the scope of the experimental analysis remains limited. The authors haven't convincingly addressed this major concern.

---

### Decision · Program_Chairs · 2026-01-26

Reject